# Associations of temperature and precipitation with malaria in children under 5: A multi-country study in Sub-Saharan Africa

Suleiman Chombo[1]*, Jovine Bachwenkizi[2], Huda Omary[3], Heavenlight A. Paulo[1], Pankras Luoga[4], Abdallah Zacharia[3], Jackline Vicent Mbishi[1], John D. Andrew[5], Isaac Y. Addo[6,7,8]

1 Department of Epidemiology and Biostatistics, Muhimbili University of Health and Allied Sciences, Dar es Salaam, Tanzania, 2 Department of Environmental and Occupational Health, Muhimbili University of Health and Allied Sciences, Dar es Salaam, Tanzania, 3 Department of Parasitology and Medical Entomology, Muhimbili University of Health and Allied Sciences, Dar es Salaam, Tanzania, 4 Department of Development Studies, Muhimbili University of Health and Allied Sciences, Dar es Salaam, Tanzania, 5 University of Galway, Galway, Ireland, 6 Cancer Institute New South Wales, Sydney, Australia, 7 Centre for Social Research in Health, University of New South Wales, Sydney, Australia, 8 The University of Sydney Susan Wakil School of Nursing and Midwifery, University of Sydney, Sydney, Australia

* sychombo@gmail.com

## Abstract

Climate change is a significant global challenge with major impacts on human health. It directly affects vector-borne diseases such as malaria by expanding vector ranges, boosting reproduction and biting rates, and shortening pathogen incubation periods. This study aimed to evaluate the association of temperature and precipitation with malaria transmission among children under five in Sub-Saharan Africa (SSA). We employed an analytical cross-sectional design to examine the relationship between temperature, precipitation, and malaria transmission among 15,009 children aged under five in six SSA countries: Burundi, Burkina Faso, Malawi, Nigeria, Tanzania, and Uganda. Historical climate data (temperature and precipitation) were retrieved from ERA-5 for the 12 months preceding the surveys. Weighted Modified Poisson regression model was used to assess the associations between climatic variables and malaria transmission. Malaria prevalence in the sample averaged 25.9%, with Nigeria (38.1%) and Burundi (38.0%) showing the highest rates. The results indicate that a one-degree Celsius rise in temperature increased malaria risk by 1.77-fold (95% CI: 1.297–2.414, p < 0.001), while a one-unit rise in squared temperature reduced risk by 1% (95% CI: 0.984–0.997, p = 0.002). Children living in regions with annual precipitation between 250–500 mm faced a 72% higher risk of malaria than those in areas receiving over 500 mm (95% CI: 1.331–2.212, p < 0.001), highlighting the nonlinear influence of climate on malaria transmission among vulnerable populations. In conclusion, the findings suggest a significant link between precipitation, temperature and increased malaria transmission in SSA. This underscores the

**Data availability statement:** Data that support the findings of this study is available upon request from Demographic Health Survey (DHS) data. Link: https://dhsprogram.com/data/dataset_admin/login_main.cfm?CFID.

**Funding:** The author(s) received no specific funding for this work.

**Competing interests:** The authors have declared that no competing interests exist.

importance of incorporating climate data into malaria control strategies to mitigate transmission risks among vulnerable populations.

## Introduction

Climate change refers to any change in climate over time, whether due to natural variability or as a result of human activity [1,2]. According to the World Health Organization (WHO), climate change is the single biggest health threat facing humanity in the 21st Century [3,4]. It is also a serious threat to ecosystems and biodiversity globally [5]. Climate change has historically occurred on Earth, but its current pace and magnitude of occurrence raise significant global concerns. Notably, global mean temperatures have increased by 1.1°C since 1900, and it is anticipated that there will be a rise in global mean temperatures of up to 5.4°C by 2100 [5,6]. Evidence shows that human activities are responsible for climate change largely due to the greenhouse gas emissions caused by burning fossil fuels, deforestation, industrialisation, harmful Agricultural practices, poor waste management practices, among others [5,7–9].

Climatic factors, such as temperature and precipitation, are known to be closely associated with the transmission of the deadly disease, malaria, as they influence mosquito breeding, parasite development, and disease spread [10,11]. Malaria, a life-threatening disease caused by Plasmodium parasites, is transmitted to humans through the bite of infected female Anopheles mosquitoes [12]. Despite being preventable and treatable, malaria continues to be a significant global public health challenge. In 2023, there were an estimated 263 million malaria cases globally (ranging from 238 to 294 million), resulting in 597,000 deaths. Out of 83 malaria-endemic countries, 29 accounted for about 95% of all malaria cases and 96% of malaria deaths worldwide. Tanzania was among the top four countries contributing to over half of all global malaria deaths, with a share of 4.3%; others were Nigeria (30.9%), the Democratic Republic of Congo (11.3%), and Niger (5.9%) [3].

Transmission of malaria is highly influenced by climatic factors such as temperature, rainfall, and humidity, as these influence mosquito distribution and longevity, the development of malaria parasites within mosquitoes, and ultimately the overall transmission of the disease [13–16]. Higher humidity and moderate temperature influence the life cycle of the parasite and mosquito by promoting their survival and development, while rainfall provides a good condition for mosquito breeding [14]. Climate change not only affects ecosystems and human health but also impacts the distribution and intensity of vector-borne diseases, including malaria [17]. Consequently, changes in these climatic factors can affect the increase or decrease of suitable habitats for malaria vectors and malaria transmission. Therefore, in the absence of attention to these changes, it is expected that there will be a change in the pattern of malaria transmission in the future [18].

Temperature, rainfall, and humidity have separate but interacting effects on malaria transmission [4,19,20]. These factors determine not only the geographical limits of the disease, but also its seasonality and intensity within those limits; thus, they influence the epidemiology of malaria, the burden of the disease, and the

efficacy of various interventions, which are the basis for designing national malaria strategic plans [3]. Evidence shows that temperature fluctuations affect parasite infection, the rate of parasite development, and mosquito biology [10]. Temperature variations can also affect gonotrophic cycle, biting rate, longevity, and maturation of mosquitoes [12,21–23]. Studies show that higher temperatures accelerate the development of the *Anopheles* mosquitoes and shorten the extrinsic incubation period of the *Plasmodium* parasite, allowing the parasite to become infectious more rapidly [21,24]. However, extreme temperatures can be detrimental to mosquito survival [25,26]. Shapiro et al. highlighted that mosquito biting rates remain low at temperatures below 15°C but reach their highest levels within the 20–30°C range [26]. Rainfall also plays a key role in creating mosquito breeding habitats, as stagnant water sources provide ideal conditions for larvae to develop, but the relationship is complex because heavy rainfall can flush away breeding habitats [27–30]. Changes in rainfall patterns, such as increased frequency of floods or droughts, can either enhance or inhibit mosquito proliferation depending on the local environment. Humidity is another essential factor, as mosquitoes are more likely to survive and transmit malaria in humid climates [31].

Due to climate change, mosquitoes have the potential to expand their geographical limit to an area that is malaria-free, leading to epidemics due to lack of natural immunity [2–3]. Rural communities in endemic regions, with limited access to healthcare and infrastructure, may experience a higher malaria burden due to climate-related changes such as floods. Moreover, climate change can affect global food systems, leading to food insecurity and malnutrition in millions of people, which makes affected populations even more vulnerable to severe malaria, lifelong complications, and early death [32,33]. Additionally, climate-related events such as floods may hinder access to healthcare by damaging roads and disrupting transportation, preventing both patients and providers from reaching health facilities [34]. These events can also disrupt the supply of essential commodities and weaken healthcare infrastructure, making it more difficult and costly to manage malaria control and elimination programs. Consequently, the overall malaria burden is likely to increase, with a particular impact on young children [3].

Despite the critical significance of climate change in public health and livelihoods, there remains a notable research gap concerning its specific impact on malaria risk, particularly among children under five in sub-Saharan Africa (SSA). While extensive studies have explored the broader implications of climate change on health, the nuanced interactions between changing climatic conditions and malaria transmission dynamics in vulnerable populations are insufficiently addressed [31,35,36]. This lack of targeted research is concerning, given that young children are particularly susceptible to malaria and its severe consequences. Understanding how key climate indicators, such as temperature and precipitation, are associated with malaria is essential for developing effective prevention strategies. This study is one of the few multi-country analyses in Sub-Saharan Africa that leverages ERA5 data in conjunction with DHS data to investigate the associations between temperature, precipitation, and malaria risk among children under five. This study provides a deeper understanding of climate-malaria interactions than country-level or ecological studies alone.

Therefore, a focused investigation into this area is important to inform public health policies and interventions aimed at protecting this at-risk demographic in the context of a changing climate. The goal of this study, therefore, is to determine how changes in temperature and precipitation are associated with malaria transmission among under five children in six highly impacted SSA countries.

## Materials and methods

### Study design and population

The study utilized an analytical cross-sectional study design to understand the associations of temperature and precipitation with malaria transmission in children under five years in Sub-Saharan Africa (SSA). We included DHS surveys from Tanzania, Nigeria, Burkina-Faso, Burundi, Malawi, and Uganda [37]. We selected these countries based on the list of the SSA countries with the highest malaria prevalence [38]. Certain countries, such as Congo and Angola, were excluded from this study due to the unavailability of either a recent DHS dataset or corresponding climatic data for the specific year of the DHS dataset.

All sub-Saharan African (SSA) countries identified as having a high malaria burden and possessing the most recent Demographic and Health Survey (DHS) data from 2015 onward were included in the analysis. SSA countries meeting the above criteria but lacking corresponding extracted climate data, specifically temperature and precipitation metrics from the ERA-5 dataset for the DHS survey year, were excluded from the study.

## Sampling technique

The Demographic and Health Survey (DHS) is a nationally representative cross-sectional survey that uses a two-stage stratified cluster sampling method to collect data on population, health, and nutrition. Clusters are selected first, followed by households within them, and information is gathered through standardized questionnaires to provide a snapshot of key indicators at a specific point in time [39]. A total of 32,490 under-fives from six SSA countries were found to be eligible for this study. We utilized an open-access health population survey, which covers more than 90 developing countries.

Permission to use this survey was obtained from the ICF Institutional Review Board, and all the methods included in this study were performed in accordance with the relevant ICF guidelines and regulations of the DHS (https://dhsprogram. com/data/dataset_admin/login_main.cfm?CFID).

## Study variables and measurements

Our study included the variable labelled as "hml35" in the DHS dataset, which represents the "positive malaria rapid diagnostic test results" as a dependent or outcome variable of interest. The variable was recoded into a binary variable, 1 as "yes", and 0 as "no". Independent variables such as the child's age in months were sub-categorized into "24 months or less" and "more than 24 months". Wealth index was recoded into three categories: poor and poorest as "poor", rich and richest as "rich", and the middle category was maintained. Mother's education variable was recoded into "no education", "primary education" as well "secondary education or higher". Precipitation was recoded into 3 categories: $0 - < 250\,mm$ as category 1, $250$–$500\,mm$ as category 2, and greater than $500\,mm$ as category 3.

Covariate variables for this study were selected based on previous literature [37,40,41]. Both continuous and categorical independent variables were included. The variables were grouped into child characteristics (child's age in months and child's sex), maternal characteristics (mother's education), interventions characteristics (Fansidar uptake during pregnancy and use of insecticide-treated nets, ITNs), household characteristics (sex of household head, wealth index, and residence type) and climatic characteristics (temperature measured in Celsius degrees and precipitation measured in millimetre (mm)). A variable, temperature, was used as continuous variables, but other variables (child's age, child's sex, residence, wealth quintile, Fansidar uptake, use of ITN and precipitation level) were used as categorical variables.

## Data analysis

**Data extraction.** We utilized household member recode (PR) and women recode (IR) datasets, as well as GPS datasets from the DHS. The cluster ID and the respective year of the country DHS were the basis for the extraction of the temperature and precipitation data. Thereafter, the climate data for temperature and precipitation for each country were combined into a single dataset, which was then merged with the respective country's DHS data. The overall dataset comprised both DHS and climate data for each country, and they were then appended into one as the Sub-Saharan DHS data with climate variables. Weighting was initially applied separately to the PR and IR datasets for each country before combining them into a single dataset. This process was repeated for each country, and the country-specific datasets were subsequently merged into one. Information on children, including sex and age, was obtained from the PR dataset, while the IR dataset was chosen to provide additional data on factors such as insecticide use, which were available in that dataset.

The study obtained a total of 42,281 children under the age of five years from the combined datasets of six SSA countries, focusing on the outcome variable of interest. Malaria proportion estimates for each sampled country were based on

this sample size. However, 27,272 participants were excluded from the study due to missing values in either of the explanatory variables used in the final model analysis. The study included the variable "use of ITNs," which was only answered by participants whose mothers previously reported having bed nets. Consequently, participants whose mothers reported not having bed nets were excluded from the study. The final sample size used for descriptive analysis of participant characteristics, and the final model analysis was 15,009.

Historical ambient temperature and precipitation data were extracted from ERA-5, a global reanalysis dataset from the European Centre for Medium-Range Weather Forecasts. A data set contains a 1-hour temporal frequency with a spatial resolution of 0.25 x 0.25 degrees (28 km x 28 km) as stipulated in the previous study [42]. We calculated the monthly mean temperature and precipitation for the 12 months preceding the survey as the key exposure window for our analysis [12]. By anchoring climatic exposure to the year prior to outcome assessment, we improve temporal alignment and enhance the interpretability of associations between environmental conditions and malaria risk among children under five [31,43]. We used zonal averaging (area-weighted averaging) to assign ERA-5-derived climate variables to DHS clusters. Specifically, for each cluster, we extracted and averaged values from all ERA5 grid cells within a defined buffer zone (10 km radius for each cluster), ensuring a more representative estimate of local climate conditions. We used this approach is appropriate for handling the spatial displacement of DHS cluster coordinates and provides more robust estimates.

These data were then matched with the geocoded addresses of DHS clusters where malaria data were collected. The climatic factors, such as temperature, were defined as the average temperature in the cluster based on the recorded temperature from January to December of the particular year at the respective cluster [12]. Likewise, precipitation was defined as the average precipitation based on recorded precipitation from January to December [31].

A *ggplot2 package* in R was used to create the density distribution plots for temperature and precipitation. Density distribution plots were generated using data from 42,281 participants, incorporating temperature and precipitation data extracted from ERA5, matched to the local environmental conditions of each participant. A heat map was utilized to visually illustrate the relationship between historical temperature and precipitation data for the corresponding years of the DHS dataset in each country and the malaria transmission. This approach highlights valuable insights into how climatic variables interact with malaria transmission across countries. The visualization aids in identifying temperature and precipitation levels susceptible to malaria outbreaks. It also serves as a tool for better understanding the influence of weather patterns on disease prevalence, contributing to more effective public health interventions.

A geographical map illustrating malaria prevalence across selected countries was generated using R, incorporating the *tmap* package.

To account for weighted analysis for the DHS dataset, a weight variable was created from the variable v005 (weight = v005/1,000,000) as per DHS guidelines [44]. Then all data were set to survey design using the "*survey*" *package of the R programming software.* Then all analyses were performed based on the survey design. For this study, a weighted modified Poisson regression model with a robust estimator was employed [45]. The modified Poisson regression model was used instead of binary logistic regression because the outcome is a common event (had a prevalence of more than 10%) [46]; binary logistic regression is suitable for rare events, but for the common events, it tends to overestimate the odds ratio and produce wider confidence intervals [47–49].

The weighted model was chosen over the unweighted version to account for the complex sampling design used during data collection, ensuring equal representation of the sampled population from various clusters.

**Variables selection.** The variable selection criteria used in this study were crosscutting. All criteria, such as "variables being included based on previous literature", "important confounding variables", "external knowledge on malaria and climate change and its impacts", as well as "p-value criteria" were considered at different angles before deciding on suitable variables for the multivariable model. The "p-value" of less than 0.2 was used as the benchmark criterion for variable selection [50].

The analysis was done using R software version 4.4.1. Covariates with p-value less than 0.05 were considered statistically significant at 5% level.

## Results and discussion

### Respondent background characteristics

Children aged 24 months or younger accounted for 5,947 (39.6%) of the sample, while females under the age of five accounted for 7,331 (48.8%). Mothers with at least a primary education level comprised 61.2% of the sample, and 71.3% reported taking Fansidar during pregnancy.

Of all study participants, 22.4% resided in urban areas. A total of 21.7% of households were in the lowest wealth quintile, 40.9% were in the middle quintile, and 37.4% were in the highest wealth quintile (see Table 1).

The findings from Fig 1 suggest that less than one-third of children under five in the six SSA countries had malaria (25.9%). Prevalence was highest in Nigeria (38.1%), Burundi (38.0%), and Malawi (36.2%), while Burkina Faso (28.0%), Uganda (19.8%), and Tanzania (7.5%) reported lower rates. This highlights malaria's persistent toll on young children in Sub-Saharan Africa.

The six Sub-Saharan countries reported to have an average annual temperature of 25 degrees Celsius (SD = 4) and median annual precipitation of 223 mm (IQR = 225).

### Precipitation distribution

The results from the density plot on precipitation indicate that most areas in the six SSA countries receive average annual precipitation ranging between 0–500 mm (See Fig 2).

### Temperature distribution

The temperature density plot revealed several clusters, indicated by multiple peaks in mean annual temperatures. This suggests that the six SSA countries experience a wide range of mean annual temperatures, from approximately 15°C to 30°C (see Fig 3). Burkina Faso emerged as the hottest among the six, while Nigeria ranked second, with mean annual temperatures between 25°C and 30°C. Burundi and Malawi displayed roughly normal temperature distributions, ranging from 17.5°C to 25°C. Tanzania and Uganda showed broader variation, with mean annual temperatures spanning 15°C to 27.5°C and multiple peaks reflecting regional differences.

### Malaria transmission and temperature

The six Sub-Saharan African (SSA) countries have an average annual temperature ranging from 20°C to 30°C, a range considered optimal for malaria transmission, thus increasing their susceptibility to the disease [50,51]. Burkina Faso has the highest average annual temperature, followed closely by Nigeria. [31,52].

Fig 4 illustrates the distribution of malaria transmission across various mean annual temperatures. In Burundi, there is increased transmission at mean annual temperatures around 20°C (ranging from 19°C to 21°C). Burkina Faso exhibits a high density of malaria transmission at mean annual temperatures between 28°C and 30°C. In Nigeria, a high density of malaria transmission is observed at mean annual temperatures above 25°C, particularly within the 26°C to 30°C range. Both Tanzania and Uganda show high malaria transmission density between 20°C and 25°C, while Malawi experiences high transmission density between 20°C and 22°C.

### Malaria transmission and precipitation

Rainfall in a given area is often linked to its altitude [53]. Regions close to sea level typically experience high temperatures and low rainfall, leading to higher malaria transmission rates [30]. Conversely, areas situated hundreds of meters above sea level, especially highland regions, generally have lower temperatures and very high rainfall, resulting in less malaria transmission [54,55].

**Table 1. Socio-demographic and Climatic factors.**

| Variables | Frequency | Percent |
|---|---|---|
| **n = 15,009** | | |
| **Child characteristics** | | |
| Child's age | | |
| <=24 months | 5947 | 39.6 |
| >24 months | 9062 | 60.4 |
| Child's sex | | |
| Male | 7678 | 51.2 |
| Female | 7331 | 48.8 |
| **Maternal characteristics** | | |
| Education of a mother | 5818 | 38.8 |
| No education | 4293 | 28.6 |
| Primary | 4898 | 32.6 |
| Secondary or higher | | |
| **Household characteristics** | | |
| Residence | | |
| Urban | 3361 | 22.4 |
| Rural | 11648 | 77.6 |
| Wealth quintile | | |
| Poor | 6139 | 21.7 |
| Middle | 3252 | 40.9 |
| Rich | 5619 | 37.4 |
| **Interventions** | | |
| Fansidar uptake | | |
| No | 4300 | 28.7 |
| Yes | 10709 | 71.3 |
| Use of (ITNs) | | |
| No | 642 | 4.3 |
| Yes | 14367 | 95.7 |
| **Climatic characteristics** | | |
| Temperature, mean (SD) | | 25 (4) |
| Precipitation, median (IQR) | | 223 (225) |
| Precipitation | | |
| <250 mm | 1783 | 11.9 |
| 250–500 mm | 8509 | 56.7 |
| >500mm | 4717 | 31.4 |
| **DHS Country Data (Year)** | | |
| Burkina Faso (2021) | 3793 | 25.3 |
| Burundi (2016−17) | 2627 | 17.5 |
| Malawi (2017) | 1207 | 8.0 |
| Nigeria (2021) | 2206 | 14.7 |
| Tanzania (2022) | 3341 | 22.3 |
| Uganda (2018−19) | 1836 | 12.2 |

## Malaria Prevalence (%) in selected SSA countries

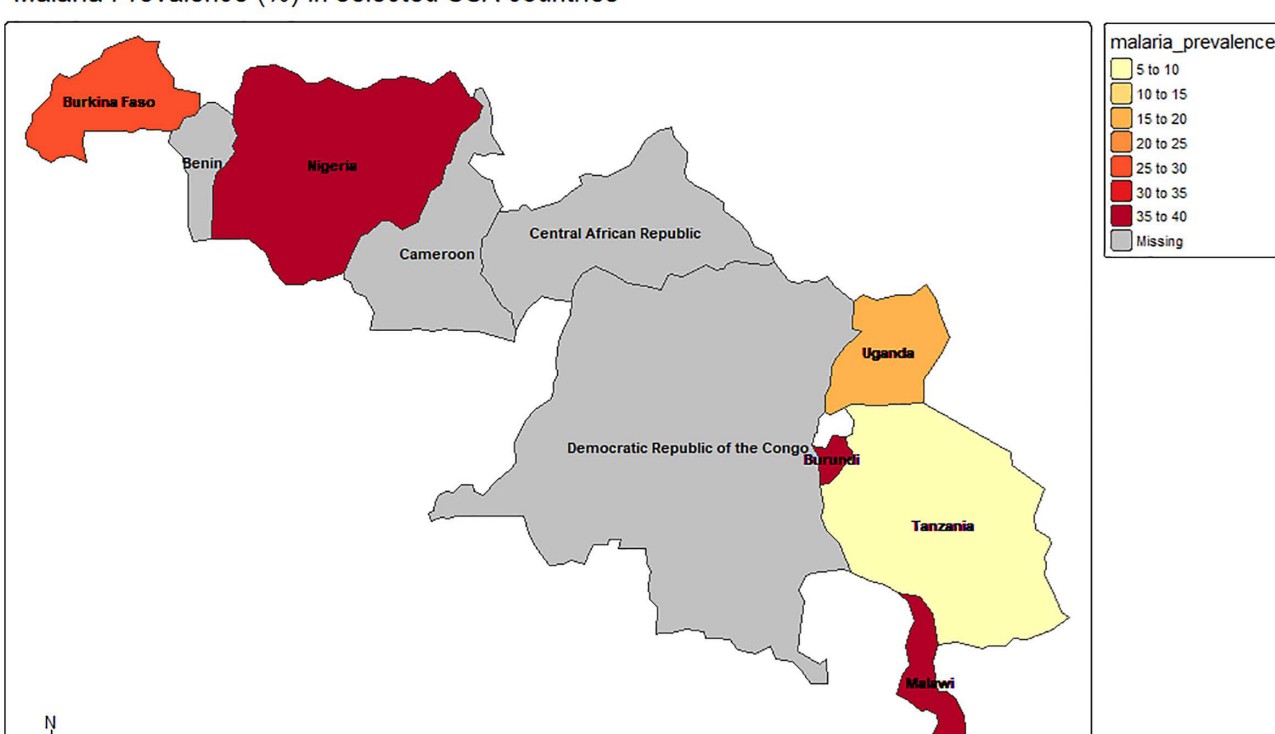

**Fig 1. Malaria prevalence among children under five by country.** The shape file used to create the map was obtained from Natural Earth (http://www.naturalearthdata.com/).

Fig 5 illustrates that, on average, regions with mean annual precipitation below 500 mm experience higher malaria transmission density across six SSA countries. In Burundi, Malawi, and Uganda, the increased malaria transmission density is most notable at precipitation levels between 250 mm and 500 mm. For Burkina Faso, the higher transmission density is predominantly observed within the 50 mm to 250 mm precipitation range. In Nigeria and Tanzania, higher malaria transmission density tends to occur at precipitation levels between approximately 50 mm and 450 mm.

The visual colour gradient implies the malaria transmission levels. Deep red areas in both Figs 4 and 5 indicate high concentrations of cases at specific temperature ranges, while lighter red bands correspond to lower case counts.

### Determinants of malaria transmission

The findings in Table 2 show that, after controlling for confounders, higher temperatures were strongly associated with increased malaria transmission among children under five in six Sub-Saharan African countries. Additionally, factors such as the child's age, mother's level of education, wealth quintile, place of residence, Fansidar uptake, as well as squared temperature and precipitation levels, were also significantly associated with higher malaria transmission rates in this age group.

Children older than 24 months had a 35% higher risk of contracting malaria compared to those aged 24 months or younger (95% CI: 1.196–1.512, p < 0.001). Children whose mothers had at least a secondary education had a 54% lower risk of malaria compared to those whose mothers had no formal education (95% CI: 0.373–0.564, p < 0.001).

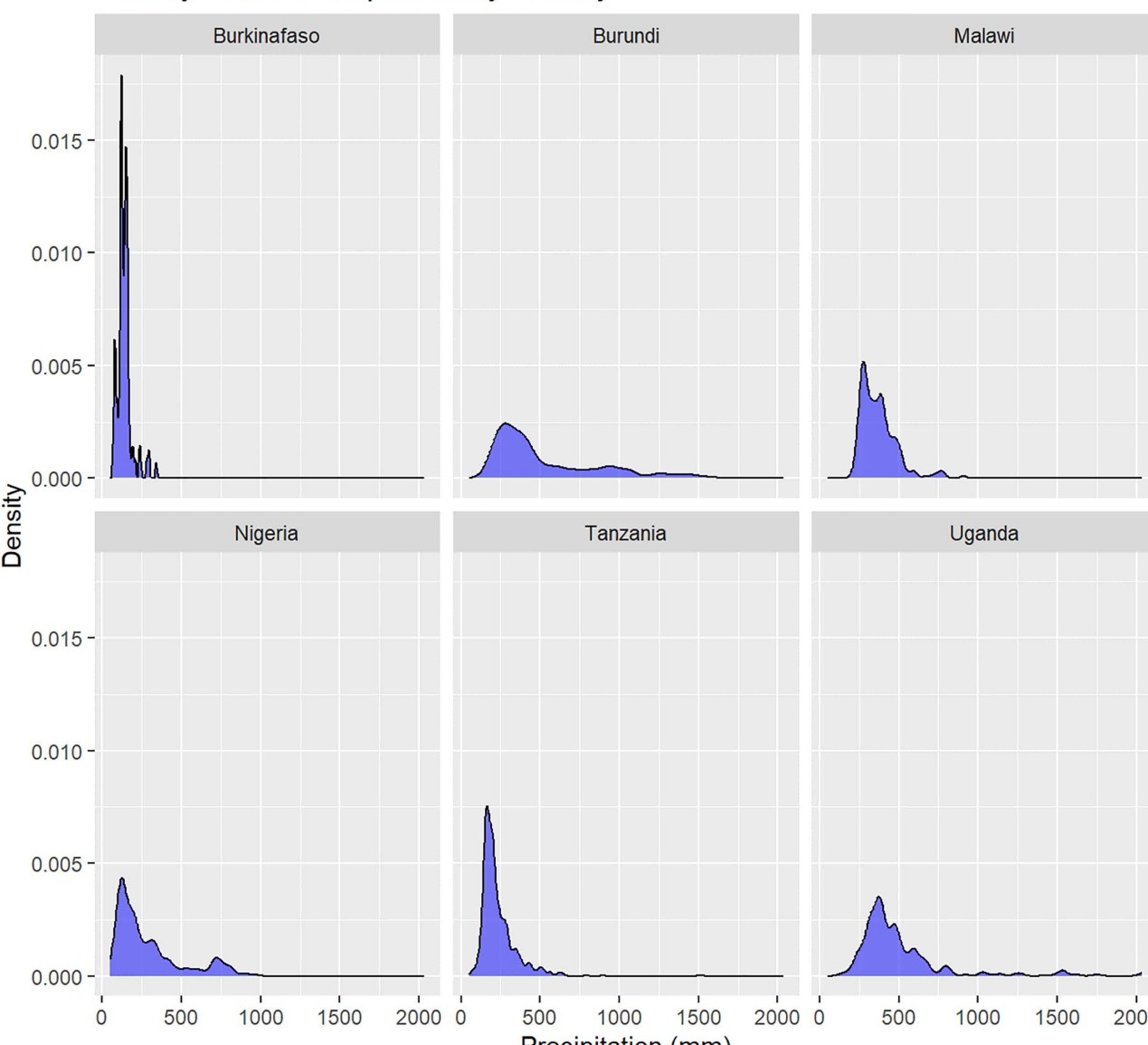

**Fig 2. Precipitation density across all six sampled SSA countries.**

Households in urban areas had a 51% lower risk of children under five contracting malaria compared to households in rural areas (95% CI: 0.380–0.626, p < 0.001). Households in the lowest wealth quintile had a 27% higher risk of malaria in children under five compared to those in the middle wealth quintile (95% CI: 1.105–1.467, p = 0.001). Conversely, households in the wealthiest quintile had a 19% lower risk of malaria in children under five compared to households in the middle wealth quintile (95% CI: 0.675–0.975, p = 0.026).

Mothers who took Fansidar during pregnancy had a 21% lower risk of their children under five contracting malaria compared to those who did not (95% CI: 0.718–0.87, p < 0.001). However, mothers who used insecticide-treated nets (ITNs)

**Fig 3. Temperature density across six sampled SSA countries.**

did not show a significant reduction in malaria risk in children under five compared to those using non-insecticide-treated nets (95% CI: 0.755–1.162, p = 0.550).

Temperature demonstrated a non-linear relationship with malaria transmission among children under five. This suggests that temperature affects malaria transmission differently at various levels. An increase of one degree Celsius was associated with 77% increased risk of malaria among children under five (95% CI: 1.297–2.414, p < 0.001). Conversely, a one-unit increase in the temperature squared resulted in a 0.99-fold increase in risks of malaria transmission among the study population (95% CI: 0.984–0.997, p = 0.002). Under five children residing in areas with mean annual precipitation ranging from 250 mm to 500 mm had 72% higher risks of having malaria compared

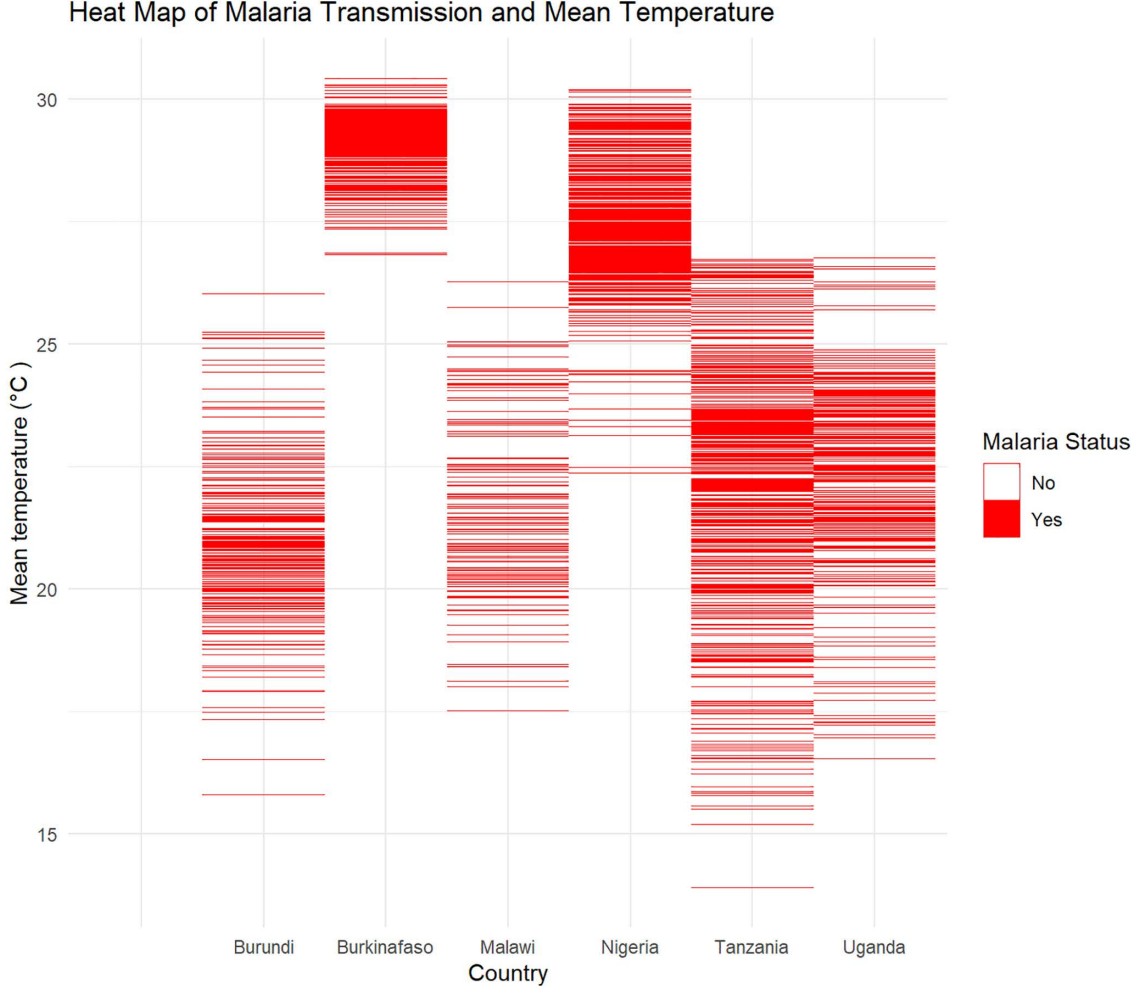

**Fig 4. Heat map of Malaria transmission and mean annual temperature.**

to those residing in areas with mean annual precipitation greater than 500 mm (95% CI: 1.331–2.212, p < 0.001). Under 5 children residing in areas with mean annual precipitation ranging between 0–250 mm indicated 31% higher risks of having malaria compared to those residing in areas with mean annual precipitation greater than 500 mm (95% CI: 0.99–1.731, P = 0.059).

## Discussions

This study primarily investigated the association of temperature and precipitation (specifically rainfall) on malaria transmission among children under five in SSA countries. While focusing on the effects of climatic factors, the study also considered other covariates to account for potential confounding influences. The results indicated an overall malaria prevalence of 25.9% across the sampled countries, highlighting the persistent public health burden of malaria. These findings are consistent with previous research on malaria prevalence in sub-Saharan Africa, including the study by Mbishi et al. [37], which reported a prevalence of 26.2% across seven SSA countries, and Anjorin et al. [40], which recorded a prevalence of 24.2% in eleven SSA countries.

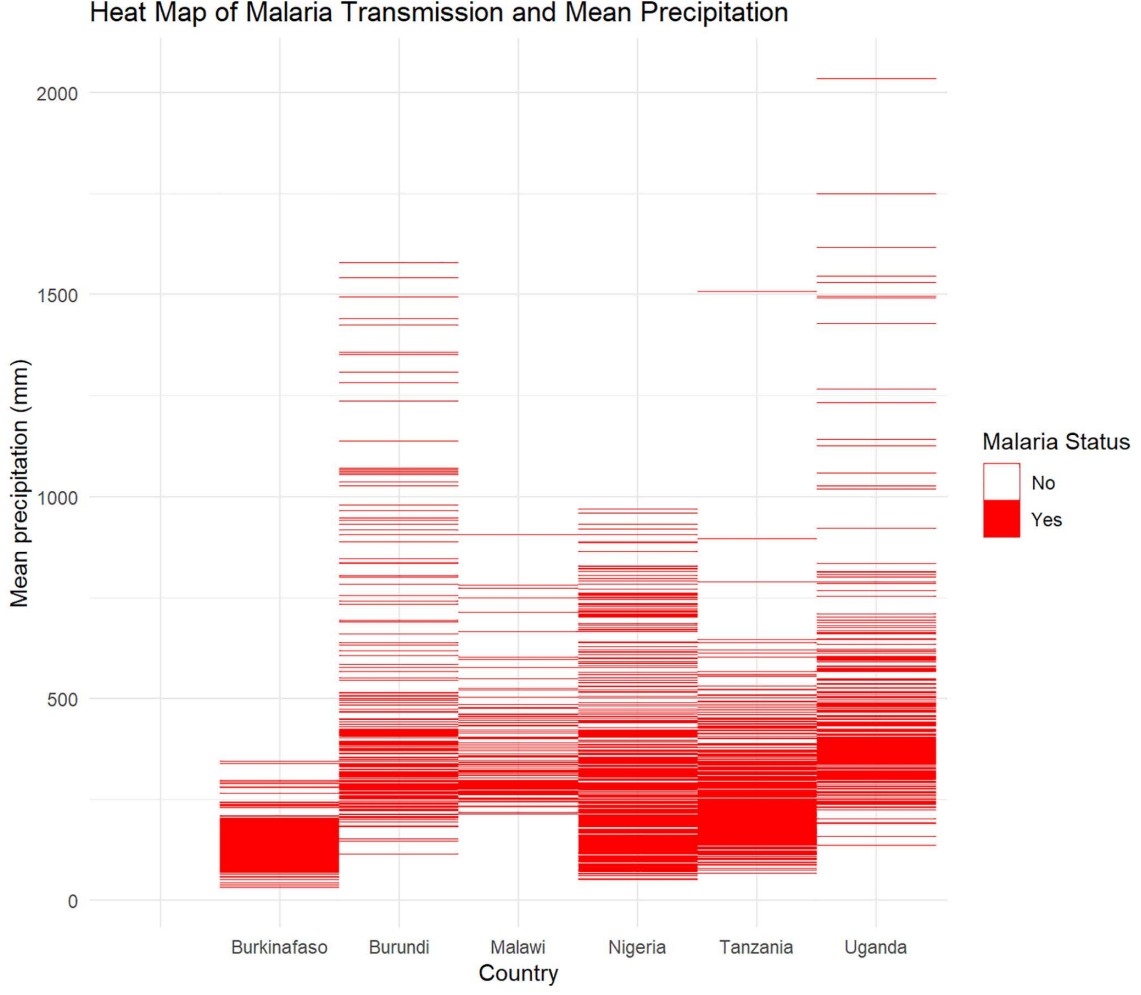

**Fig 5. Heat map of Malaria transmission and mean annual precipitation.**

 

The results highlight the multifaceted nature of malaria transmission and the need for integrated control strategies that incorporate both environmental and socio-economic determinants. Our findings demonstrate that climatic factors, such as temperature and precipitation, play a significant role in malaria transmission across the sampled SSA countries, reinforcing the well-established link between climatic conditions and vector-borne disease dynamics. Beyond climatic factors, several socio-demographic and intervention-related variables, such as a child's age, maternal education level, wealth quintile, residence, and Fansidar uptake were significantly associated with malaria prevalence. The findings demonstrate that temperature plays a crucial role in influencing malaria transmission among children under five years of age. This effect can be attributed to several temperature-dependent factors that directly or indirectly influence malaria transmission in this vulnerable age group [12]. Warmer temperatures accelerate the life cycle of *Anopheles* mosquitoes, leading to higher vector density [22]. Higher temperatures also increase the number of blood meals taken and the number of eggs laid by the mosquitoes, which increases the number of mosquitoes in a given area [56]. Warmer temperatures help mosquitoes to feed frequently and increase the frequency of blood feeding and more bites, increasing the exposure of children to malaria [35]. Higher temperatures speed up the extrinsic incubation period (EIP) of *Plasmodium* parasites inside mosquitoes [57].

**Table 2. Association of climatic and socio-demographic factors with malaria transmission.**

| Variables | Univariable model | | Multivariable model | |
|---|---|---|---|---|
| | cRR (95% Conf. Int) | P-value | aRR (95% Conf. Int) | P-value |
| Child's age | | | | |
| <=24 months | 1 | | 1 | |
| >24 months | 1.388 (1.228-1.568) | <0.001 | 1.345 (1.196-1.512) | <0.001 |
| Child's sex | | | | |
| Male | 0.958 (0.896-1.024) | 0.210 | 0.974 (0.914-1.038) | 0.415 |
| Female | 1 | | 1 | |
| Education of a mother | | | | |
| No education | 1 | | 1 | |
| Primary | 0.736 (0.638-0.849) | <0.001 | 0.889 (0.778-1.016) | 0.084 |
| Secondary or higher | 0.304 (0.249-0.37) | <0.001 | 0.458 (0.373-0.564) | <0.001 |
| Residence | | | | |
| Urban | 0.31 (0.243-0.396) | <0.001 | 0.488 (0.38-0.626) | <0.001 |
| Rural | 1 | | 1 | |
| Wealth quintile | | | | |
| Poor | 1.446 (1.244-1.681) | <0.001 | 1.274 (1.105-1.467) | 0.001 |
| Middle | 1 | | 1 | |
| Rich | 0.548 (0.456-0.658) | <0.001 | 0.811 (0.675-0.975) | 0.026 |
| Fansidar uptake | | | | |
| No | 1 | | 1 | |
| Yes | 0.831 (0.743-0.929) | 0.001 | 0.790 (0.718-0.87) | <0.001 |
| Use of insecticide-treated nets (ITNs) | | | | |
| No | 1 | | 1 | |
| Yes | 1.123 (0.848-1.489) | 0.418 | 0.936 (0.755-1.162) | 0.550 |
| Temperature | 1.472 (1.077-2.012) | 0.015 | 1.770 (1.297-2.414) | <0.001 |
| Temperature squared | 0.993 (0.987-0.999) | 0.040 | 0.990 (0.984-0.997) | 0.002 |
| Precipitation | | | | |
| 0-<250 mm | 1.905 (1.456-2.493) | <0.001 | 1.309 (0.99-1.731) | 0.059 |
| 250–500 mm | 1.865 (1.426-2.439) | <0.001 | 1.717 (1.331-2.212) | <0.001 |
| >500 mm | 1 | | 1 | |

cRR, crude relative risk; aRR, adjusted relative risk 1-Reference group.

This means mosquitoes become infectious more quickly, increasing malaria transmission risk. In warm conditions, children may spend more time outdoors, especially in the evenings when mosquito activity is highest, increasing their risk of getting bitten [58]. During warmer nights, caregivers and children may avoid sleeping under ITNs due to discomfort, leading to increased exposure to mosquito bites [59]. In warm conditions, people may dress in lighter, shorter clothing, which can increase skin exposure and the risk of mosquito bites [35].

Conversely, the study suggests that at very high temperatures, malaria risk does not continue to rise indefinitely but rather slows down or declines slightly [22]. This may be due to the reduced mosquito survival at both extremely high and low temperatures. While moderate temperature increases mosquito activity, excessive heat can be lethal to mosquitoes, reducing their lifespan and reproductive capacity, and excessive cold reduces the metabolic processes of mosquitoes and makes them less capable of seeking blood meals [60], hence reducing malaria transmission. Our findings are similar to several previous studies [12,20,55,61]

Rainfall (precipitation) is another environmental factor that has been shown to play a significant role in malaria transmission. The results from this study indicate a significant effect of rainfall on malaria transmission in SSA countries. Increased rainfall creates temporary pools, puddles, and slow-moving streams that serve as ideal breeding habitats. In areas with poor drainage or where water accumulates, mosquito larvae thrive, leading to higher adult mosquito populations [62]. High precipitation levels increase humidity, which extends the lifespan of adult mosquitoes [63]. Longer survival increases their chances of becoming infected with *Plasmodium* parasites and transmitting malaria to humans. Heavy rainfall can damage housing structures, forcing people to seek shelter in conditions with poor mosquito protection, such as open houses, temporary shelters [64]. Flooding may displace populations, leading to overcrowded living conditions where malaria transmission risk is increased [65–68]. The findings of this study correspond with the findings of Okiring et al. [15] and Liu et al. [11], which showed that malaria transmission was higher at lower levels of precipitation compared to higher levels.

Regarding socio-demographic characteristics, our results indicate that children who are older than 24 months are at a higher risk of malaria infection compared to those aged 24 months or younger. A similar trend has been observed in studies conducted in Uganda [69], Nigeria [70], Malawi [71], and Ethiopia [72], highlighting a consistent trend in different malaria transmission settings. This increased risk may be due to several factors, including the reduced use of ITNs as caregivers tend to prioritise younger siblings, leaving older children more exposed to mosquito bites [73]. Younger children typically receive more consistent care from mothers and caregivers, ensuring they sleep under treated bed nets more frequently compared to older children [74]. Additionally, children under 24 months are generally less mobile and tend to remain indoors, limiting their exposure to mosquito habitats, while older children are more likely to engage in outdoor play, especially in the early evening and morning when mosquitoes are active, thus increasing the risk of mosquito contact [75]. Furthermore, as the children grow, the naturally acquired antibodies decline, making children vulnerable to the infection [70].

Our findings corroborate those documented by previous studies [37,40,41] in the SSA region, which highlight maternal education as a key predictor of malaria infection among children under five. We found that mothers with primary education or higher education were associated with lower malaria transmission to their under-five children compared to those with no formal education. This underlines the role of education in public health concerns. Mothers who have attended school are likely to possess formal knowledge and good practice that help safeguard their children's well-being. More importantly, mothers with secondary or higher education were associated with significantly lower malaria risks to their under-five children compared to those with primary education and those with no formal education. This could be because these mothers have a better understanding of malaria-related issues, having been exposed to various forums, seminars, and training on malaria prevention. Education also has the potential to improve household income, as women with higher education are more likely to secure better-paying jobs, enabling them to afford malaria prevention tools like ITNs, insecticides, or to build homes that prevent mosquito entry. Additionally, educated mothers are more empowered to seek timely healthcare for their children when needed [76,77].

An educated mother is more likely to understand the risks associated with malaria transmission to children under five and to know various control measures to protect them, such as using bed nets, maintaining clean surroundings, and dressing children in long sleeves and trousers [37]. In contrast, lack of education may translate to poor knowledge, which might lead to delay in seeking healthcare and facilitate late detection of malaria, which might lead to the increased transmission to other family members and the risk of complications [37,40,78]. Therefore, strengthen girls' and women's education remains a critical upstream stratergy.

Intermittent Preventive Treatment in pregnancy (IPTp) is one of the key strategies for malaria prevention and control for pregnant mothers and their unborn child. The intervention is meant to prevent a pregnant mother from having malaria, which might be transferred to an unborn child. According to this study, the Fansidar uptake during pregnancy as an IPTp approach had a significant impact in reducing malaria transmission to children under-five. The results imply that mothers

had a lower risk of contracting malaria and hence reduced the risk for the child to have malaria infection. The uptake of Fansidar during pregnancy may also imply the behavioral pattern of a mother to be willing to take whatever available and feasible measures to prevent her child from malaria, as well as taking measures immediately whenever she notes alarming signs or symptoms in her child. The findings of this study also concur with other literature, which reported the significant impact of Fansidar uptake or SP towards malaria prevention to under-five [37,78].

The use of insecticide-treated mosquito nets (ITNs) is widely recognized as a highly effective intervention for reducing malaria transmission, as supported by extensive literature [79,80] Interestingly, although the prevalence was lower among ITN users, the association was not statistically significant. significant. The lack of statistical significance observed in this study may be attributed to several factors. First, the predominance of plank houses in many rural areas, often characterized by structural crevices, facilitates mosquito entry and creates microenvironments that enhance mosquito survival and increase human-vector contact [81]. Second, behavioral factors such as outdoor evening and early morning activities expose children to mosquito bites during peak biting hours before net use [75]. Additionally, growing evidence of insecticide resistance among local mosquito populations may further threaten the protective efficacy of ITNs [82,83]. Moreover, the use of worn-out or damaged ITNs with holes may compromise their protective efficacy, allowing mosquitoes to reach individuals even when nets are used [84]. These factors combined may help explain the discrepancy between the expected and observed impact of ITNs in this setting.

Household wealth status impacts malaria transmission is well published [37,40,85,86] and well evident in this study. Children under five from middle and high-income households have a lower risk of malaria transmission compared to those from poor households. Poor wealth status often means limited access to healthcare and resources, as well as inadequate nutrition and poor housing quality, leading to increased exposure to mosquito bites and a higher risk of malaria infection.

Malaria has been referred to as a disease of poverty that disproportionately affects poor people residing in rural areas. In this study, children from urban areas accounted for 51% lower risks of malaria compared to children residing in rural settings. Urban residents have better access to health services, while rural residents face barriers in accessing healthcare services, which might lead them to have poor health-seeking behavior. Furthermore, environmental factors such as wetlands, farming activities, and housing conditions may put them at risk of increased mosquito bites. These findings are consistent with other studies, which also highlight the disparity in malaria prevalence between rural and urban areas in SSA. Improving health infrastructure, enhancing access to healthcare services, and upgrading housing conditions in rural areas are crucial steps in reducing malaria transmission among children under five [37,87,88].

Behavioural factors related to a child's sex can significantly influence the risk of malaria infection. Based on behavioural differences, it was hypothesized that boys might have a higher risk of malaria infection compared to girls, as boys are often observed spending more time playing outdoors. However, the findings of this study revealed that male children had a lower risk of malaria infection compared to their female counterparts. These results align with a study conducted in Ethiopia [89], which also reported a higher infection rate among female children than male children. One possible explanation could be, caregivers tend to prioritize male children over female children when seeking healthcare, including malaria testing and treatment. This gender-based disparity in care-seeking behavior has been documented in various studies [46,90,91]

Policy and practice implications integrated with study findings. National malaria control programmes should incorporate climate information into routine planning by establishing early warning systems that use real-time temperature and rainfall data to identify high-risk periods and locations. Strengthened collaboration between health, meteorological, and environmental agencies is essential to ensure that vector control measures are deployed proactively in response to changing climatic conditions. Priority should be given to areas experiencing rising temperatures or intermediate rainfall levels, where the risk is highest. At the same time, investment in climate-resilient health infrastructure and capacity building for local health workers will be critical to sustain malaria services during climate-driven surges. These environmental strategies must also be complemented by socio-economic measures, including expanding girls' education, improving rural housing

and drainage systems, and supporting consistent uptake of preventive interventions such as IPTp. Together, these integrated actions provide a direct bridge to the study's conclusion.

## Strengths and limitations of the study

The primary strength of this study was the use of nationally representative data with a large sample size, across six SSA countries, ensuring robust statistical power and generalizability of the findings to countries with similar demographic and climatic conditions. The study integrated a combination of data from publicly available sources – DHS and ERA-5, which increases the reliability of the data. The study considered a 12-month window of climatic exposure, capturing seasonal variations in temperature and precipitation. Additionally, the study used a weighted modified Poisson regression model to help account for the complex survey design and potential biases, improving the accuracy of the results.

However, this study has several limitations. Other environmental variables that could influence malaria transmission, but were not considered in this study, include humidity, which affects mosquito survival and parasite development, and wind speed, which can influence mosquito dispersal. Vegetation and soil moisture are important as they determine the availability of mosquito breeding sites, while elevation can reduce transmission at higher altitudes due to cooler temperatures. Additionally, proximity to water bodies, which serve as breeding sites for mosquitoes, could also play a significant role in transmission dynamics. Including these factors would offer a more comprehensive understanding of environmental influences on malaria. Also, the cross-sectional design limits the ability to establish causality between climatic factors and malaria transmission. Potential confounding variables, such as healthcare access and socio-economic factors, may not have been fully accounted for. The use of historical climate data from ERA-5 may have inaccuracies, particularly in rural areas, and the study's reliance on self-reported malaria cases could lead to underreporting. Additionally, the 12-month climate data may not capture seasonal variations or delayed effects on malaria transmission. The findings may not be fully generalisable beyond the six SSA countries included, and other environmental factors influencing transmission were not considered.

Additionally, while the literature consistently demonstrates that insecticide-treated mosquito nets significantly reduce malaria transmission, our study interestingly found no statistically significant association for this intervention. One possible explanation is the reliance on self-reported data for ITN use, which may be subject to recall or social desirability bias, leading to misclassification. Additionally, our analysis was cross-sectional and may not fully capture seasonal variations in both ITN use and malaria transmission. It is also possible that high community-level ITN coverage in some areas reduced measurable differences between users and non-users, or that factors such as net age, physical condition, and proper use were not adequately captured in the dataset. Finally, variations in vector resistance to insecticides across the study settings could attenuate the effectiveness of ITNs.

We recommend that future research prioritize longitudinal and cohort study designs to establish causal relationships and to better elucidate the delayed effects of climatic factors on malaria transmission.

## Conclusion

The study findings reveal a significant association between precipitation, temperature, and increased malaria transmission in SSA countries. This highlights the necessity of integrating environmental factors into malaria control strategies to minimize transmission risks among vulnerable populations. Additionally, the study emphasizes the role of other contributing factors, including maternal education, area of residence, wealth quintile of the household and the age of a child as well as malaria interventions such as Fansidar uptake during pregnancy and ITN use. These findings advocate targeted malaria control strategies that address these risk factors alongside protective interventions.

## Author contributions

**Conceptualization:** Suleiman Chombo.

**Data curation:** Suleiman Chombo, Jovine Bachwenkizi.

**Formal analysis:** Suleiman Chombo, Jovine Bachwenkizi.

**Methodology:** Suleiman Chombo.

**Software:** Suleiman Chombo.

**Validation:** Suleiman Chombo.

**Visualization:** Suleiman Chombo.

**Writing – original draft:** Suleiman Chombo, Jovine Bachwenkizi, Huda Omary.

**Writing – review & editing:** Suleiman Chombo, Jovine Bachwenkizi, Huda Omary, Heavenlight A. Paulo, Pankras Luoga, Abdallah Zacharia, Jackline Vicent Mbishi, John D. Andrew, Isaac Y. Addo.

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
