## [Decision Letter · Decision Letter 0]

18 Jul 2025

Dear Dr. Chombo,

Thank you for submitting your manuscript to PLOS ONE. After careful consideration, we feel that it has merit but does not fully meet PLOS ONE’s publication criteria as it currently stands. Therefore, we invite you to submit a revised version of the manuscript that addresses the points raised during the review process.

Thank you for submitting your manuscript to PLoS ONE. After careful consideration, we felt that your study has the potential to be published if it is revised to address specific topics raised by the reviewers. According to the reviewers, there are several specific topics where further improvements would be of substantial benefit to readers; for example, the methods  section appears to be limited and  should be revised. For your guidance, a copy of the reviewers' comments was included below.  

We look forward to receiving your revised manuscript.

Kind regards,

Luzia H Carvalho, Ph.D.

Academic Editor

PLOS ONE

Journal Requirements:

2. In the online submission form you indicate that your data is not available for proprietary reasons and have provided a contact point for accessing this data. Please note that your current contact point is a co-author on this manuscript. According to our Data Policy, the contact point must not be an author on the manuscript and must be an institutional contact, ideally not an individual. Please revise your data statement to a non-author institutional point of contact, such as a data access or ethics committee, and send this to us via return email. Please also include contact information for the third party organization, and please include the full citation of where the data can be found.

3. Please amend the manuscript submission data (via Edit Submission) to include author Suleiman Chombo.

4. Please amend your authorship list in your manuscript file to include author Suleiman University of Health and Allied Chombo.

Reviewers' comments:

Reviewer's Responses to Questions

**Comments to the Author**

1. Is the manuscript technically sound, and do the data support the conclusions?

Reviewer #1: Yes

Reviewer #2: Partly

Reviewer #3: Partly

2. Has the statistical analysis been performed appropriately and rigorously?

Reviewer #1: Yes

Reviewer #2: No

Reviewer #3: Yes

3. Have the authors made all data underlying the findings in their manuscript fully available?

Reviewer #1: Yes

Reviewer #2: Yes

Reviewer #3: No

4. Is the manuscript presented in an intelligible fashion and written in standard English?

Reviewer #1: Yes

Reviewer #2: No

Reviewer #3: Yes

Reviewer #1: This study investigates the association between temperature, precipitation, and malaria transmission among children under five in six Sub-Saharan African (SSA) countries. The research is highly relevant, given the increasing concerns about climate change and its impact on vector-borne diseases. The study utilizes an analytical cross-sectional design, incorporating both survey data and historical climate data from ERA-5, and applies a Weighted Modified Poisson regression model to assess associations. The findings highlight a significant relationship between climatic factors and malaria prevalence, particularly emphasizing how increased temperature and specific precipitation levels elevate the risk of malaria transmission. These insights contribute to the growing body of evidence advocating for climate-informed malaria control strategies. The study's methodological approach, use of weighted analysis, and incorporation of climate variables strengthen its reliability and applicability in public health planning.

Before proceeding with publication, I recommend making some minor revisions.

Abstract

Ensure consistency in decimal formatting throughout the abstract and manuscript (e.g., using either one or two decimal places uniformly).

Introduction

Line 110-113: "Although numerous studies have investigated the overall health impacts of climate change, the specific interplay between shifting climatic conditions and malaria transmission dynamics in vulnerable populations remains underexplored ." To strengthen this statement, you should cite relevant studies that examine both the general health effects of climate change and those that focus on malaria transmission dynamics.

Methods

The study design should be clearly stated. The Demographic and Health Survey (DHS) follows a two-stage stratified cluster sampling design, and this should be properly described in the Study Design and Population section.

The authors should specify how sample weights were implemented in the data analysis section. Applying sample weights is crucial to ensure that the findings are representative of the target population.

Result

No comments

Discussion

Line 340-42: Remove this "These findings reinforce the consistently high malaria prevalence in SSA and

highlight the need for continued surveillance and intervention efforts to reduce the disease

burden among children under five."

Line 348-50: Remove this" These results highlight the multifaceted nature of malaria

349 transmission and the need for integrated control strategies that incorporate both environmental

350 and socio-economic determinants."

A 'Policy Recommendations' section should be added before the Conclusion to provide actionable insights based on the study’s key findings. This section should highlight policy implications and suggest evidence-based interventions.

Reviewer #2: Thank you for submitting this manuscript entitled “Associations of Temperature and Precipitation with Malaria Transmission among Children Under Five in Six Sub-Saharan African Countries: A Cross-Sectional Study.” This study addresses an important and timely topic. The manuscript is based on a large sample size, uses publicly available and reputable data sources (DHS and ERA5), and applies appropriate statistical methods (weighted modified Poisson regression) to assess associations between climate factors and malaria prevalence. The results have meaningful public health implications. However, there are several areas where the manuscript should be substantially improved before it can be considered for publication.

Title and Abstract

1. Lines 1–2: The current title is overly long. Consider simplifying to: “Climatic Factors and Malaria Risk among Under-Five Children in Sub-Saharan Africa: A Multi-Country Analysis.”

2. Lines 20–46: The abstract contains awkward phrasing. For example, “a 1.77-fold in risks” should be revised to “a 1.77-fold increase in risk,” and “0.99-fold in risks” to “a 1% decrease in risk.” The overall writing should be polished for conciseness and clarity.

Introduction (Lines 47–120)

3. Lines 58–70: Some cited studies are outdated or unclear (e.g., “(12–14)”). Please incorporate up-to-date and globally relevant literature, to enhance the discussion of climate impacts on mosquito-borne disease dynamics.

4. Lines 109–117: Please highlight the novelty of your study. For example, emphasize that this is one of the few multi-country SSA studies combining ERA-5 climate data with DHS microdata to examine under-five malaria risk.

Methods (Lines 121–217)

5.Lines 142–150: Clarify that “hml35” refers to positive malaria rapid diagnostic test results. Reinforce the definition of the exposure window (12 months prior to survey) for temperature and precipitation to strengthen the logic of temporal association.

6.Lines 194–200: The methodology for generating heat maps should be better described. How were transmission densities calculated across clusters?

7. Line 214: The use of a p-value threshold (0.2) alone for variable selection is not robust. Please consider using AIC/BIC or stepwise regression for a more defensible selection process.

Results (Lines 218–324)

8.Lines 227–236: Clarify whether the “Overall” prevalence in Figure 1 is weighted across countries.

9.Lines 244–253: Figures 2 and 3 would benefit from clearer x-axis labels and indication of temperature or precipitation range across clusters. Consider labeling country groups directly in density curves.

10.Lines 313–317: Since a non-linear temperature effect is observed, consider using spline regression or a GAM to model the dose–response more flexibly, and illustrate it graphically.

Discussion (Lines 331–462)

11.Lines 348–368: The mechanisms linking temperature to mosquito behavior and malaria transmission are over-detailed and speculative in parts (e.g., “light clothing increases bite exposure”). Please ensure all statements are backed by references.

12.Lines 377–390: The precipitation mechanism discussion is somewhat repetitive. Consider summarizing in a schematic figure or flowchart.

13.Lines 394–400: Expand the explanation of how age affects risk, possibly with more references. For example, differences in exposure behavior, immunity, or bed net prioritization. Moreover, what is “add citation” in line 397.

14.Lines 456–462: The sex-based difference in malaria risk is briefly mentioned. Since your findings are contrary to common expectations, please provide a more detailed interpretation or reference evidence to support the observation.

Conclusion (Lines 488–501)

15.Lines 497–500: The recommendation for future research could be better structured. Suggest explicitly that longitudinal and cohort studies are needed to establish causality and clarify lagged climate–malaria effects.

Others

16.Figures 1–5: Improve figure legends, label axes, and standardize the color scales used in heat maps. A clear legend indicating levels of malaria risk would help.

17.Table 2 : Clearly specify reference categories in footnotes (e.g., “Secondary education vs. no education”).

18. Some references are incomplete or inconsistently formatted. Please use the reference style required by the journal and update key references.

Reviewer #3: 1. The manuscript title, main objective, and conclusions are consistent.

The data sources are valid (DHS, ERRA-5). However, details must be detailed in the methodology to better judge the validity of the results and facilitate the reproducibility of analysis:

a. Reference year of the DHS of the countries;

b. inclusion criteria used by the DHS to include the case of malaria. Did these criteria differ according to the countries?

c. Precision of the spatial scale (cluster) of meteorological data collection in the countries;

d. note on the method of calculating certain variables such as temperature density.

2. It is recognized in the literature that the use of insecticide-treated mosquito nets has a significant impact on malaria transmission. It would be worth discussing the possible limitations leading to the insignificance of this intervention in this study.

**Do you want your identity to be public for this peer review?** For information about this choice, including consent withdrawal, please see our Privacy Policy

Reviewer #1: No

Reviewer #2: **Yes: ** Haoran Wang

Reviewer #3: No

---

## [Author Response · Author response to Decision Letter 1]

15 Aug 2025

Response to reviewer 1

RESPONSE MATRIX

COMMENT RESPONSE

Ensure consistency in decimal formatting throughout the abstract and manuscript (e.g., using either one or two decimal places uniformly). Thank you for this feedback. For the estimates and values for confidence intervals, we used 3 decimal places, and one decimal place for percentages, as some of the results such as p-value <0.001, reporting it as <0.01 could be misleading.

Line 110-113: "Although numerous studies have investigated the overall health impacts of climate change, the specific interplay between shifting climatic conditions and malaria transmission dynamics in vulnerable populations remains underexplored." To strengthen this statement, you should cite relevant studies that examine both the general health effects of climate change and those that focus on malaria transmission dynamics Citations added.

The study design should be clearly stated. The Demographic and Health Survey (DHS) follows a two-stage stratified cluster sampling design, and this should be properly described in the Study Design and Population section.

The authors should specify how sample weights were implemented in the data analysis section. Applying sample weights is crucial to ensure that the findings are representative of the target population. A clear study design was added, and sampling procedure was clearly stated.

Sampling weight was clearly specified.

Line 340-42: Remove this "These findings reinforce the consistently high malaria prevalence in SSA and highlight the need for continued surveillance and intervention efforts to reduce the disease burden among children under five The pre-mentioned lines were included to preamble the discussion part. We thought putting these statements would add value to the discussion part.

Line 348-50: Remove this" These results highlight the multifaceted nature of malaria transmission and the need for integrated control strategies that incorporate both environmental and socio-economic determinants." The statements were removed.

A Policy Recommendations' section should be added before the Conclusion to provide actionable insights based on the study’s key findings. This section should highlight policy implications and suggest evidence-based interventions Policy recommendations added after the conclusion.

Response to reviewer 3

Comment Response

Add reference year of the used country respective DHS. Reference years were added in Table 1 under the DHS country Data (Year)

inclusion criteria used by the DHS to include the case of malaria. Did these criteria differ according to the countries? Inclusion and exclusion criteria were added.

The inclusion and exclusion criteria did not differ according to the country.

Precision of the spatial scale (cluster) of meteorological data collection in the countries;

Thank you for your valuable comment. In this study, we used ERA-5 reanalysis data, which provides gridded meteorological variables at approximately 0.25° × 0.25° (~28 km × 28 km) spatial resolution. To integrate these data with DHS cluster locations, we employed spatial interpolation (or spatial averaging) to assign ERA5-derived climate variables (temperature and precipitation) to each DHS cluster based on its GPS coordinates.

However, we have updated our manuscript by adding the following information to clarify our approach:

“We used zonal averaging (area-weighted averaging) to assign ERA-5 derived climate variables to DHS clusters. Specifically, for each cluster, we extracted and averaged values from all ERA5 grid cells within a defined buffer zone (10 km radius for each cluster), ensuring a more representative estimate of local climate conditions. We used this approach is appropriate for handling the spatial displacement of DHS cluster coordinates and provides more robust estimates”.

note on the method of calculating certain variables such as temperature density A note was taken care of.

It is recognized in the literature that the use of insecticide-treated mosquito nets has a significant impact on malaria transmission. It would be worth discussing the possible limitations leading to the insignificance of this intervention in this study One possible explanation is the reliance on self-reported data for ITN use, which may be subject to recall or social desirability bias, leading to misclassification. Additionally, our analysis was cross-sectional and may not fully capture seasonal variations in both ITN use and malaria transmission. It is also possible that high community-level ITN coverage in some areas reduced measurable differences between users and non-users, or that factors such as net age, physical condition, and proper use were not adequately captured in the dataset. Finally, variations in vector resistance to insecticides across the study settings could attenuate the effectiveness of ITNs. We have now included these points in the discussion to acknowledge these limitations and provide context for the observed results.

Response to reviewer 2

Comment Response

Line 214: The use of a p-value threshold (0.2) alone for variable selection is not robust. Please consider using AIC/BIC or stepwise regression for a more defensible selection process. Thank you for the input. We acknowledge that AIC and BIC are commonly applied for model selection when comparing multiple competing models to determine the best fit. However, in our study, a p-value threshold of less than 0.2 was employed specifically for variable selection—transferring variables from the univariable model to the multivariable model. This approach was not intended for overall model selection but rather served as a criterion for identifying relevant predictors for multivariable analysis.

Lines 227–236: Clarify whether the “Overall” prevalence in Figure 1 is weighted across countries. Thank you regarding this input.

All analyses performed in the study were weighted as have been highlighted in the methodology section.

Lines 244–253: Figures 2 and 3 would benefit from clearer x-axis labels and indication of temperature or precipitation range across clusters. Consider labeling country groups directly in density curves. Thank you again for this input.

Figures 2 and 3 were revised to take into account the country groupings.

Lines 313–317: Since a non-linear temperature effect is observed, consider using spline regression or a GAM to model the dose–response more flexibly, and illustrate it graphically.

Discussion (Lines 331–462)

Thank you for the input.

The inclusion of a squared temperature term in the model effectively captured the nonlinear relationship between temperature and malaria, aiding in model linearization. Given that this transformation yielded satisfactory model performance, we did not find it necessary to pursue more complex modeling techniques such as spline regression and GAM.

17.Table 2 : Clearly specify reference categories in footnotes (e.g., “Secondary education vs. no education”).

Thank you for the input.

All reference categories presented in Table 2 of the originally submitted manuscript were explicitly identified by assigning the code “1” to the respective reference group in each categorical variable. The footnote was added to clearly define the reference group under Table 2.

16.Figures 1–5: Improve figure legends, label axes, and standardize the color scales used in heat maps. A clear legend indicating levels of malaria risk would help. Thank you for the input.

We considered the option you suggested, but it could not be implemented in that form. Since temperature and precipitation were treated as continuous variables in the heat maps, deriving discrete malaria risk levels proved methodologically problematic. Therefore, we opted to retain the original figure format.

In the heat maps, the red coloration reflects the density of reported malaria cases. Deep red areas indicate high concentrations of cases at specific temperature ranges, while lighter red bands correspond to lower case counts. This visual gradient preserves the continuous nature of the underlying climatic variables and aligns with the original analytical approach.

15.Lines 497–500: The recommendation for future research could be better structured. Suggest explicitly that longitudinal and cohort studies are needed to establish causality and clarify lagged climate–malaria effects.

Others Thank you for the suggestion.

The explicit recommendation based on longitudinal study for establishing causality have now been added.

Lines 348–368: The mechanisms linking temperature to mosquito behavior and malaria transmission are over-detailed and speculative in parts (e.g., “light clothing increases bite exposure”). Please ensure all statements are backed by references. Thank you for the input.

Redundant statements have been removed and citations have been added where necessary.

Lines 142–150: Clarify that “hml35” refers to positive malaria rapid diagnostic test results. Thank you for the input.

Hml35 was re-defined as “positive malaria rapid diagnostic test results” as suggested.

Lines 20–46: The abstract contains awkward phrasing. For example, “a 1.77-fold in risks” should be revised to “a 1.77-fold increase in risk,” and “0.99-fold in risks” to “a 1% decrease in risk.” The overall writing should be polished for conciseness and clarity.

Introduction (Lines 47–120 Thank you. The entire results of the Abstract has been rewritten for clarity.

Reinforce the definition of the exposure window (12 months prior to survey) for temperature and precipitation to strengthen the logic of temporal association Thank you.

Reinforcement statement was added.

Lines 109–117: Please highlight the novelty of your study. For example, emphasize that this is one of the few multi-countries SSA studies combining ERA-5 climate data with DHS microdata to examine under-five malaria risk. Thank you for the suggestion.

Statements were added to highlight the novelty of the study.

Lines 1–2: The current title is overly long. Consider simplifying to: “Climatic Factors and Malaria Risk among Under-Five Children in Sub-Saharan Africa: A Multi-Country Analysis.”

The title was revised.

---

## [Decision Letter · Decision Letter 1]

24 Sep 2025

Please submit your revised manuscript by Nov 08 2025 11:59PM. If you will need more time than this to complete your revisions, please reply to this message or contact the journal office at plosone@plos.org . A rebuttal letter that responds to each point raised by the academic editor and reviewer(s). You should upload this letter as a separate file labeled 'Response to Reviewers'.A marked-up copy of your manuscript that highlights changes made to the original version. You should upload this as a separate file labeled 'Revised Manuscript with Track Changes'.An unmarked version of your revised paper without tracked changes. You should upload this as a separate file labeled 'Manuscript'.

We look forward to receiving your revised manuscript.

Kind regards,

Luzia H Carvalho, Ph.D.

Academic Editor

PLOS ONE

Journal Requirements:

Reviewers' comments:

Reviewer's Responses to Questions

**Comments to the Author**

Reviewer #1: All comments have been addressed

Reviewer #2: (No Response)

Reviewer #3: All comments have been addressed

2. Is the manuscript technically sound, and do the data support the conclusions?

Reviewer #1: Yes

Reviewer #2: Partly

Reviewer #3: Yes

3. Has the statistical analysis been performed appropriately and rigorously?

Reviewer #1: Yes

Reviewer #2: Yes

Reviewer #3: Yes

4. Have the authors made all data underlying the findings in their manuscript fully available?

Reviewer #1: Yes

Reviewer #2: Yes

Reviewer #3: Yes

5. Is the manuscript presented in an intelligible fashion and written in standard English?

Reviewer #1: Yes

Reviewer #2: Yes

Reviewer #3: Yes

Reviewer #1: The manuscript titled "Temperature and Precipitation as Drivers of Malaria in Children Under 5: A Multi-Country Study in Sub-Saharan Africa" is ready for publication.

Reviewer #2: Thank you for the revised manuscript. The quality has improved, and several of my earlier comments have been addressed. However, a number of issues remain and require further revision

1.The title currently reads “Children Under5”. Please correct the spacing to “Children Under 5.

2.The word “drivers” (Line 1) suggests causality, which is not appropriate for a cross-sectional study. A more neutral term such as “associations” or “relationships” would be preferable.

Suggested title:

“Associations of Temperature and Precipitation with Malaria in Children Under 5: A Multi-Country Study in Sub-Saharan Africa.”

3.Lines 20–45: The abstract remains relatively long. Consider condensing it to approximately 300 words�according to the requirements of the journal�, focusing on the study objective, main findings, and implications rather than detailed statistics.

4.Line 246: The use of p<0.2 as a criterion for variable inclusion lacks sufficient justification. Consider adding a sensitivity analysis (e.g., with p<0.1) to show robustness, or cite relevant methodological literature.

5.Lines 227–236: The explanation of weighting remains brief. Please include the DHS weighting formula in the main text (e.g., weight = v005/1,000,000) and reference DHS methodological documentation, not only supplementary notes.

6.Lines 259–263: The narrative duplicates what is shown in Figure 1. A more concise statement is recommended, such as: “Prevalence varied considerably, from below 10% in Tanzania to over 30% in Nigeria and Burundi (Figure 1).

7.Some passages are repetitive, for example, Lines 379–384 and 388–397 both emphasize the importance of temperature for malaria transmission. Consider merging and streamlining these sentences to improve clarity.

8.The policy recommendations starting at Line 562 appear disconnected from the conclusion. I suggest integrating them into the final Discussion paragraph so they flow naturally into the Conclusion.

Reviewer #3: (No Response)

**Do you want your identity to be public for this peer review?** For information about this choice, including consent withdrawal, please see our Privacy Policy

Reviewer #1: No

Reviewer #2: **Yes: ** Haoran Wang

Reviewer #3: No

---

## [Author Response · Author response to Decision Letter 2]

3 Oct 2025

The title currently reads “Children Under5”. Please correct the spacing to “Children Under 5. Thank you.

The spacing was corrected. 2

The word “drivers” (Line 1) suggests causality, which is not appropriate for a cross-sectional study. A more neutral term such as “associations” or “relationships” would be preferable. Thank you for the comment.

The word “drivers” was removed, and the title was revised on the reviewer’s suggestion. 1

Lines 20–45: The abstract remains relatively long. Consider condensing it to approximately 300 words according to the requirements of the journal, focusing on the study objective, main findings, and implications rather than detailed statistics Thank you for the comment.

The abstract has been revised to 263 words. 36-53

Lines 227–236: The explanation of weighting remains brief. Please include the DHS weighting formula in the main text (e.g., weight = v005/1,000,000) and reference DHS methodological documentation, not only supplementary notes. Thank you for the comment.

The weighting formula was included in the previously submitted manuscript. 242

Lines 259–263: The narrative duplicates what is shown in Figure 1. A more concise statement is recommended, such as: “Prevalence varied considerably, from below 10% in Tanzania to over 30% in Nigeria and Burundi (Figure 1). Thank you for your comment. The narrative reports the computed prevalence for each of the six countries, while Figure 1 illustrates the overall prevalence range across the sample. We consider it important to keep precise country-specific estimates rather than relying only on the broader range shown in the figure. Nevertheless, we have revised the text to be more concise in line with your suggestion

Some passages are repetitive, for example, Lines 379–384 and 388–397 both emphasize the importance of temperature for malaria transmission. Consider merging and streamlining these sentences to improve clarity. Thank you for the comment. The sentences have been restructured to enhance clarity and eliminate redundancy. 324

The policy recommendations starting at Line 562 appear disconnected from the conclusion. I suggest integrating them into the final Discussion paragraph so they flow naturally into the Conclusion Appreciated for the comment. We revised the recommendations to align with the findings in the discussion and conclusion as suggested by the reviewer. 578-600

Line 246: The use of p<0.2 as a criterion for variable inclusion lacks sufficient justification. Consider adding a sensitivity analysis (e.g., with p<0.1) to show robustness, or cite relevant methodological literature. Citation supporting the use of the criterion was added 258 &

259

---

## [Editor Report · Decision Letter 2]

7 Oct 2025

Associations of Temperature and Precipitation with Malaria in Children Under 5: A Multi-Country Study in Sub-Saharan Africa

PONE-D-25-15162R2

Dear Dr. Chombo,

We’re pleased to inform you that your manuscript has been judged scientifically suitable for publication and will be formally accepted for publication once it meets all outstanding technical requirements.

Kind regards,

Luzia H Carvalho, Ph.D.

Academic Editor

PLOS ONE
---

## [Editor Report · Acceptance letter]

PONE-D-25-15162R2

PLOS ONE

Dear Dr. Chombo,

I'm pleased to inform you that your manuscript has been deemed suitable for publication in PLOS ONE. Congratulations! Your manuscript is now being handed over to our production team.

Kind regards,

on behalf of

Dr. Luzia H Carvalho

Academic Editor

PLOS ONE